# Foliar Functional Traits of Resource Island-Forming Nurse Tree Species from a Semi-Arid Ecosystem of La Guajira, Colombia

**DOI:** 10.3390/plants11131723

**Published:** 2022-06-29

**Authors:** Gabriela Toro-Tobón, Fagua Alvarez-Flórez, Hernán D. Mariño-Blanco, Luz M. Melgarejo

**Affiliations:** Laboratorio de Fisiología y Bioquímica Vegetal, Departamento de Biología, Facultad de Ciencias, Universidad Nacional de Colombia—Sede Bogotá, Carrera 45 #26-85, Bogotá 111321, Colombia; gtorot@unal.edu.co (G.T.-T.); hmarino@unal.edu.co (H.D.M.-B.)

**Keywords:** nurse trees, leaf physiological traits, chlorophyll a fluorescence, pigments, leaf anatomy

## Abstract

Semi-arid environments characterized by low rainfall are subject to soil desertification processes. These environments have heterogeneous landscapes with patches of vegetation known as resource islands that are generated by nurse species that delay the desertification process because they increase the availability of water and nutrients in the soil. The study aimed to characterize some foliar physiological, biochemical, and anatomical traits of three nurse tree species that form resource islands in the semi-arid environment of La Guajira, Colombia, i.e., *Haematoxylum brasiletto, Pithecellobium dulce*, and *Pereskia guamacho*. The results showed that *H. brasiletto* and *P. dulce* have sclerophyllous strategies, are thin (0.2 and 0.23 mm, respectively), and have a high leaf dry matter content (364.8 and 437.47 mg/g). Moreover, both species have a high photochemical performance, reaching Fv/Fm values of 0.84 and 0.82 and PI_ABS_ values of 5.84 and 4.42, respectively. These results agree with the OJIP curves and JIP parameters. Both species had a compact leaf with a similar dorsiventral mesophyll. On the other hand, *P. guamacho* has a typical succulent, equifacial leaf with a 97.78% relative water content and 0.81 mm thickness. This species had the lowest Fv/Fm (0.73) and PI_ABS_ (1.16) values and OJIP curve but had the highest energy dissipation value (DIo/RC).

## 1. Introduction

Semi-arid and arid environments cover about 40% of the Earth’s land surface and are expanding and intensifying as a result of climate change [1,2]. These environments have ecological and environmental problems resulting from vulnerability to water deficit, soil desertification, and frequent, extreme weather [3,4], which affect the productivity and physiological, biochemical, and anatomical responses of plants [5].

In Colombia, semi-arid and arid ecosystems cover 17% of the surface, where the Department of La Guajira stands out, located in the Caribbean and classified as a pre-Caribbean dry belt with two large biomes: tropical desert and tropical dry forest [6]. Alta Guajira-Colombia has an average annual rainfall between 218 and 532 mm, with a bimodal distribution that has a pronounced peak in November and a lower peak in June. The mean annual temperature is 26 °C, with a mean monthly variation of ~3.0 °C and an evapotranspiration of 2000 mm [7], generating constant water deficits [8].

In these semi-arid environments, vegetation is distributed in patches known as resource islands [9,10,11]. These islands are developed thanks to ecological facilitation processes that are mediated by nurse species, which have unique characteristics that allow them to create improved understory conditions that buffer soil temperature, stabilize pH, increase water availability through the accumulation of organic matter and hydraulic lifting, and promote the availability and cycling of nutrients through the accumulation of leaf litter and increased decomposition [11,12,13]. As a result, the process of facilitation begins below the nurse plant’s canopy, enhancing the survival, growth, and fitness of other plants, increasing biodiversity and forming a resource island. Consequently, resource islands fulfill structural and support functions in semi-arid ecosystems, reduce wind erosion, prevent nutrient loss and desertification, and conserve biodiversity by providing resources to wild species and regenerating native species [11,14].

Since climate change is expected to increase drought periods in semi-arid ecosystems, it is important to understand the structural and functional adaptation strategies of resource-island-forming nurse plants in Colombia [15,16]. Therefore, various authors have proposed using foliar functional traits, such as relative water content, leaf area, leaf thickness, leaf dry mass content, stomatal density, stomatal conductance, proline content, and total sugar content, to elucidate the responses of and effects on vegetation in this ecosystem [2,17,18,19,20]. In general, vegetation in semi-arid environments uses different strategies to tolerate droughts that vary depending on the species and scales of measurement [1]. Plants respond to water deficit by modifying its leaf area [21], increasing the specific leaf area, and decreasing leaf thickness [22]. Some species perform strict stomatal control to prevent water loss and increase water use efficiency, which result in a high relative water content in the leaves [22,23,24].

Under drought conditions, in vivo analysis of chlorophyll a fluorescence has been used to understand the physiological responses of plants because of the high sensitivity to alterations and changes in PSII performance [25]. This analysis also evaluates the state of the photosynthetic apparatus under stressful conditions, such as high temperatures [26], high light incidence [27], and water stress [28]. Likewise, the JIP test can be used, which provides information on the behavior (structure, conformation, and function) of the photosynthetic apparatus in any physiological state of plants [29,30,31].

Furthermore, plants synthesize and accumulate compatible osmolytes in the cytoplasm at the biochemical level in semi-arid environments to avoid cellular dehydration, such as soluble sugars and proline. These osmolytes play an osmoprotective role, maintaining the fluidity of plasma membranes and the proper activity of enzymes [32]. The content of photosynthetic pigments varies according to the availability of water and nutrients [33,34], generating changes in the photosynthetic rates of species in semi-arid environments. It has been observed that carotenoids play an important role in the adaptation of species to water deficits and to high light intensities since they are involved in mechanisms of excess energy dissipation to avoid photoinhibition of the photosynthetic system [29,33]. Additionally, foliar anatomical characteristics are used to evaluate the ability of species to adapt to different environments [17]. In semi-arid environments, some species have a large number of trichomes on the leaf surfaces as well as a cuticle to avoid water loss and create a barrier against the environment [35,36]. Modifications have also been observed in the thickness and arrangement of mesophyll tissues to decrease space, improve the diffusion of CO_2_, increase foliar photosynthetic capacity, and increase water retention in tissues [37,38,39].

Previous taxonomic studies in La Alta Guajira [40,41,42] observed that the floristic composition of resource islands corresponds to the *Opuntia caracasanae*-*Prosopietea juliflorae* class, which contains most resource island forming and understory species. The predominant species were *Opuntia caracasana* and *Prosopis juliflora* [43]. Other abundant species included *Haematoxylum brasiletto*, *Pithecellobium dulce*, and *Pereskia guamacho*. The objective of this study was to characterize the physiological, biochemical, and anatomical characteristics of *H. brasiletto*, *P. dulce*, and *P. guamacho* leaves in order to elucidate some of the functional strategies that these species possess to develop these ecosystems and form resource islands.

## 2. Results

### 2.1. Physiological Traits in Leaves

The physiological characterization with the functional traits (Table 1) evidenced that *H. brasiletto* had the smallest, thinnest leaves with an intermediate leaf dry mass content (LDMC) and the lowest relative water content (RWC). On the other hand, *P. guamacho*, a species with succulent leaves, had thicker leaves with a greater volume storing high water contents, where water content (WC) and RWC were higher, while the LDMC and water saturation deficit (WSD) were lower. The *P. dulce* leaves had a similar leaf area (LA) and specific leaf area (SLA) to that of *P. guamacho* and the highest LDMC. The other foliar traits (WC, RWC, WSD, and thickness) were between the characteristics of *H. brasiletto* and *P. guamacho*.

For stomatal conductance (g_s_), *H. brasiletto* and *P. guamacho* had higher values in the predawn hours (3:30–5:30 h) than around noon and the twilight hours (17:30–19:30 h). While *P. dulce* had the highest g_s_ at noon, indicating that this species does not perform stomatal closure. For stomatal density (SD), *P. dulce* had the highest value, followed by *H. brasiletto* and *P. guamacho*. Both *P. dulce* and *H. brasiletto* are hypostomatic species, while *P. guamacho* is an amphistomatic species.

The spider diagrams (Figure 1) and Table 2 show the parameters obtained from the rapid phase of the chlorophyll a fluorescence emission for the three nurse species. PI_ABS_ and Φ_Po_, also known as Fv/Fm, were similar in *H. brasiletto*, and *P. dulce. P. guamacho* had lower values, which indicated some type of photochemical alteration that prevented a greater maximum quantum efficiency of PSII. For quantum efficiencies Φ_Eo_ and Ψ_o_, *H. brasiletto* had the highest values, followed by *P. dulce* and *P. guamacho*. In the direct energy flows (ABS/RC, DIo/RC, and TRo/RC), *H. brasiletto* and *P. dulce* had similar, low values, indicating a high number of open reaction centers (RC) and a high flow of electrons along the chain. Despite this, *P. guamacho* had higher values for ABS/RC, DIo/RC, and TRo/RC; that is, there had more closed RC. However, for the parameters ETo/RC and REo/RC, *P. guamacho* had similar values to the other two species. In phenomenological energy flows, *P. guamacho* stood out with high values for DIo/CSo, which indicated that a large part of the energy dissipated in the form of heat. Opposite to direct flows, the high values for ABS/CSo, TRo/CSo, ETo/CSo, and REo/CSo observed in *H. brasiletto* and *P. dulce* showed a greater photosynthetic capacity of the electron transport chain for processing the energy in the light pulse.

The OJIP curve (Figure 2) evidenced a similar performance between the three species in the photochemical phase O-J (2 ms). In the thermal phases of the curve (J-I at 30 ms, I-P at 300 ms), *P. guamacho* had lower peaks for I and P than the other two species because it gradually decreased the intensity of fluorescence, reaching the lowest maximum fluorescence (Fm) and smallest area below the curve (Figure 2). On the other hand, *P. dulce* presented the highest Fm value (P peak) followed by *H. brasiletto*.

### 2.2. Foliar Biochemical Traits

*P. guamacho* presented the lowest contents of chlorophyll a, chlorophyll b, total chlorophylls, and carotenoids and the lowest Chla/Chlb ratio (Figure 3). There were no differences between *H. brasiletto* and *P. dulce* in these photosynthetic pigments. *H. brasiletto* had the highest proline content, followed by *P. dulce* and *P. guamacho* (Figure 3F). There were no differences between the three species for the total sugar content (Figure 3G). 

### 2.3. Foliar Anatomical Traits

*H. brasiletto* has paripinnately compound leaves, with ovate leaflets that have an emarginated cleft at the apex (Figure 4A) and uniseriate trichomes on both surfaces. The leaf epidermis has curved anticline walls on the adaxial and abaxial sides (Figure 4B,C). Anisocytic stomata (three cells of similar size enclose the guard cells) were only observed on the abaxial side (Figure 4C). The edge of the stomata had a narrow lip shape, with a protruding surface, regular margin and wide aperture field (Figure 4C). A dorsiventral mesophile was observed (Figure 4D) in the cross-section of the leaves with a uniseriate epidermis and a single layer of palisade parenchyma. There were vascular bundles immersed in the spongy parenchyma along the lamina. The midrib of the leaves (Figure 4E,F) had several short and parallel rows of xylem, representing 27.613 ± 3.517% of the width of the midrib; the vessels were angular to circular with diameters between 10 μm and 14 μm, and the walls of the vessels were 2 μm wide (Table 3). The phloem was evident under the xylem on the abaxial side. The two vascular tissues were surrounded by a layer of sclerenchyma (fibers).

*P. dulce* has a bigeminate leaf, with ovate leaflets that have an emarginated cleft at the apex (Figure 5A). The leaf epidermis had sinuous anticlinal walls on the adaxial and abaxial sides (Figure 5B,C). Paracytic stomata (1 to 2 cells adjacent to the guard cells, with a longitudinal axis parallel to the longitudinal axis of the guard cells) were only observed on the abaxial side. The edge of the stomata had a bat shape (according to terminology from Trofimov and Rohwer (2018)), with a protruding surface, regular margin, and wide aperture field (Figure 5C). A dorsiventral mesophile was observed in the cross-section of the leaf blade (Figure 5D), with a thick cuticle covering the epidermis on both sides, uniseriate epidermis, a single layer of palisade parenchyma, and vascular bundles immersed in the spongy parenchyma. The phloem was located in the adaxial and abaxial zones of the leaves and, in the adaxial zone, had two parts (Figure 5E–F). In the abaxial zone, the phloem formed a thick stripe. In addition, the xylem tissue extended in rows, representing 19.203 ± 7.631% of the width of the midrib, with vessel diameters between 10 μm and 11 μm and vessel walls 2 μm wide (Table 3). The vascular tissues were surrounded by a thick strip of sclerenchyma (fibers). 

*P. guamacho* has simple and succulent leaves, with an obtuse shape and a cuspid apex (Figure 6A). The leaf epidermis had curved anticline walls on the adaxial and abaxial sides of the leaves (Figure 6B,C). *P. guamacho* has anphystomatic leaves and paracytic stomata, with a narrow circular edge that protrudes, but it is interrupted at both ends of the stoma opening field (Figure 6C). A uniseriate epidermis and an equifacial mesophile were observed in the cross-section of the leaf blade (Figure 6D), with small vascular bundles distributed along the lamina. The vascular tissues were arranged in the midrib (Figure 6E,F) in two small diameter bundles. The vessels of the xylem had a diameter between 7 μm and 15 μm, with walls between 2 μm and 4 μm wide, representing 43.023 ± 12.075% of the width of the midrib (Table 3). The leaf blade had calcium oxalate drusen that varied in shape and size (Figure 6D–F).

The measurements of the tissues of the leaf blade and the midrib in the leaf cross-sections (Table 3) showed that the upper epidermis was thicker in *P. guamacho*, followed by *H. brasiletto* and *P. dulce*. No differences in thickness were found between the palisade parenchyma but there were differences between *H. brasiletto* and *P. dulce* for the spongy one. *P. guamacho* had a greater width of the inner wall of the xylem vessels, and H. *brasiletto* had a greater length of the xylem in the midrib. Finally, the high percentage represented by xylem with respect to the width of the central rib in the *P. guamacho* leaves was notable, followed by *H. brasiletto* and *P. dulce*.

## 3. Discussion

In the characterization of the functional leaf traits, *H. brasiletto* and *P. dulce* had similar strategies to deal with the water limitations of the environment; both species have thinner leaves, higher dry mass content, and greater water saturation deficits than *P. guamacho.* The leaf thickness values for *H. brasiletto* and *P. dulce* were similar to the average reported for deciduous species [44] and for species from semi-arid areas [17]. A thin leaf corresponds to a denser and thinner mesophyll, as observed in the anatomical measurements, which does not accumulate large amounts of water (low WC and RWC) and does not have a high foliar maintenance cost [2]. Similarly, a dense leaf indicates that there is a high inversion of dry matter per unit of leaf area that intercepts light [18,44], which decreases the conductance of the mesophyll and affects gas exchange and photosynthetic rate [45].

*P. guamacho*, unlike *H. brasiletto* and *P. dulce*, had a greater leaf thickness, a lower dry mass content, and higher RWC and WC values, which are characteristic of drought-tolerant succulent species with leaves that store high water contents [22,43,44]. These characteristics were observed in the leaf anatomy, along with an equifacial parenchyma with large vacuoles that stored water [37,46,47].

Leaf area is related to the water balance of plants [48], and the low values in *H. brasiletto* and *P. dulce* demonstrated a strategy for minimizing water loss because a smaller area means a smaller transpiration area. Similar results were reported by Guerra and Scremin-Dias [17] for trees in semi-arid environments. Notably, *H. brasiletto* and *P. dulce* have compound leaves with a better exchange of air on the surface, improving the efficiency of heat transfer in environments with higher temperatures [49]. *P. guamacho* leaves have a greater leaf area, are thick, and have a low surface area ratio and low volume (SA:V), with a maximum volume for water storage, a minimum surface for the loss of water by transpiration [37], fewer stomata per mm^2^, and lower g_s_ (Table 1).

The SLA values observed in the three species are common in environments with low nutrient availability that are prone to drought [22,50]. Similar results were found by Chaturvedi et al. [44] for species in deciduous tropical forests, but they are higher than those reported by Guerra and Scremin-Dias [17] for trees in semi-arid environments and lower than those recorded by De Souza et al. [24] in deciduous species in a tropical dry forest. On the other hand, the foliar pH of the three species was similar to that reported for woody species from semi-arid and dry sub-humid regions [51]. The ranges recorded for the three nurse tree species corresponded to a tendency towards a basic pH because, in semi-arid environments, some species use stomatal control through the production of abscisic acid (ABA), which increases the intracellular accumulation of Ca^2+^ and the alkalinization of guard cells, increasing foliar pH [51]. The foliar pH indicated that these species invest less in defenses and produce a better quality leaf litter [52], denoting *H. brasiletto*, *P. dulce*, and *P. guamacho* as nurse trees.

The g_s_ for *H. brasiletto* and *P. guamacho* at noon indicated partial stomatal closure, a strategy to reduce water loss, restrict leaf transpiration, and make better efficient use of water [23,53]. This stomatal control has been observed in tropical dry forest species [53], in deciduous plants in semi-arid regions [54], in desert species [55], and in succulent species [37,46]. On the other hand, *P. dulce* did not decrease g_s_ at noon, which probably led to an increase in the photosynthetic rate because of the entry of permanent CO_2_ to the leaf [49], posing a hydraulic risk for the plants since there is a greater water demand where plants are more prone to the cavitation of the xylem and embolisms [23,56].

The higher stomatal density in *H. brasiletto* and *P. dulce* than in *P. guamacho* allowed them to better regulate leaf transpiration and make efficient use of water [18,57]. SD values similar to those of *P. dulce* were reported by Guerra and Scremin-Dias [17] and Yücedağ et al. [58] in species in semi-arid environments. Values similar to *H. brasiletto* are observed in species of deciduous trees [59] and shrubs and species in semi-humid environments [58]. In contrast, *P. guamacho* had a low SD that was characteristic of succulent species [46,59] and used as a conservative strategy for water use [37].

The fluorescence analyses of chlorophyll a showed that *P. guamacho* had the lowest values for Φ_Po_ (Fv/Fm) and PI_ABS_, indicating low levels of electron transfer or heat dissipation strategies. This is consistent with the reports by Aragón-Gastélum et al. [60] and Jardim et al. [61] who indicated that succulent species in Cactaceae and Agavaceae families have low levels of ETR and high levels of NPQ that contribute to tolerance to adverse factors in semi-arid environments. Similarly, *P. guamacho* had the highest values for ABS/RC, DIo/RC, TRo/RC, ETo/RC, REo/RC, and DIo/CSo, which indicated a possible inhibition in the transfer of electrons through the accumulation of excess light energy, with possible effects on the reoxidation of the electron donor, resulting in the accumulation of P680^+^ (PSII RC) [27,28], or a possible reduction in the activity of PSII active reaction centers via the inactivation of the oxygen evolving complex (OEC) [62]. These results show the possible strategies that *P. guamacho* uses to dissipate excess excitation energy through thermal dissipation to maintain the energy balance between the flow of absorbed energy and the transport of electrons to protect the photosynthetic system, especially PSII, from photo-oxidative damage [63,64]. Therefore, *P. guamacho* had low quantum efficiencies (Φ_Po_, Φ_Eo_, and Ψ_o_), a low PI_ABS_, and lower fluorescence values at points J, I, and P than the other two species.

*H. brasiletto* and *P. dulce* had high quantum efficiencies and a high PI_ABS_, which are common in unstressed plants [26,65]. The low values in the direct flow parameters showed that these two species had more active CR, that not all electron acceptors (quinone pool) were reduced, and that there was a high rate of Q_A_ reoxidation [27]. On the other hand, the values of the phenomenological parameters (per CS) in *H. brasiletto* and *P. dulce* indicated that these species have high average values of photons absorbed by molecules of the antenna complex (ABS/CSo), high values of trapped energy (TRo/CSo), which translate into a high transport of electrons (ETo/CSo), and a high reduction in the final electron acceptors on the side of PS I (REo/Cso) [66]. That is, these two species efficiently use light energy for the transport of electrons in the photosynthetic process since they have high Φ_Po_, Φ_Eo_, and Ψ_o_, and an increase in the fluorescence of chlorophyll a with a transient tendency that is typical of OJIP [27,67].

In the characterization of the foliar biochemical traits, it was observed that *H. brasiletto*, and *P. dulce* presented higher contents of Chl_a_, Chl_b_, Chl_total_, and C than *P. guamacho*, which may be a possible cause of the better performance in quantum efficiency in these two species since chlorophyll is key to light harvesting and biochemical energy production in the photosynthetic process [29,68,69], and carotenoids play a special role in protecting PSII from photooxidative damage, allowing the dissipation of PSII energy as heat [28,29,70]. For photosynthetic pigments, a positive correlation has been observed between chlorophyll content and chlorophyll a fluorescence parameters [71]. The data found in the nurse tree species were similar to those reported by Chaturvedi et al. [44] in the Chl content for tropical deciduous forest species but higher than those reported in the photosynthetic pigment content for semi-arid species [72].

As observed in the results, *H. brasiletto* had the highest proline content in the leaves (Figure 3F), which was possibly a drought tolerance strategy since proline is an amino acid that fulfills multiple functions as an osmoregulator, stabilizer of proteins and membranes, and eliminator of reactive oxygen species (ROS) [26,28,73]. The content of total sugars showed that the concentrations in the three species maintain the metabolic activity of cells [74], with values similar to those reported for species in semi-arid areas [75].

The anatomical characterization showed that *H. brasiletto* has unicellular trichomes and a hypostomatic distribution of stomata, similar traits were reported by Lersten et al. [76] for the *Caesalpinieae* tribe and by Martel et al. [77] for *Caesalpinia spinosa*. Similarly, *P. dulce* has a hypostomatic leaf with paracytic stomata. These traits were also observed by García and Torrecilla [78] and Alvarado et al. [79]. Furthermore, *H. brasiletto* and *P. dulce* leaves had a similar mesophyll structure, with a dorsiventral organization and a compact mesophilic structure that corresponded to thin leaves typical of sclerophyll leaves adapted to semi-arid environments [17,80,81]. The presence of collenchyma under the epidermis in the middle veins provided support, structure, mechanical strength and flexibility [35,36]. Likewise, the presence of sclerenchyma reduces wilting and provides support, rigidity, and stability to the tissue, preventing cell collapse during periods of drought [35,36,39]. Alvarado et al. [79] also reported sclerenchyma in *P. dulce* leaves. On the other hand, the presence of cuticle, as identified in *P. dulce*, provides a barrier to avoid the loss of excessive water via transpiration and damage from a high incidence of sunlight [82,83]. This strategy can reduce the loss of water generated by stomatal opening during the hours with a high vapor pressure deficit. The sinuosity observed in the walls of the adaxial and abaxial epidermis of *P. dulce* is another strategy that increases cell stiffness, which prevents cell collapse in drought conditions [39]. This sinuosity was observed also by García and Torrecilla [78].

*P. guamacho* had the foliar anatomy that is typical of succulent species, as described by Bailey [84], Mauseth and Landrum [85], Duarte and Hayashi [86], and Maciel et al. [87]. Its leaf has a homogeneous mesophyll that allows the accumulation of water as a strategy to survive drought conditions, where cells are arranged with minimal intercellular spaces [37,38,88,89]. The presence of calcium oxalate crystals in the form of radially oriented drusen of different shapes and sizes has also been reported by Mauseth and Landrum [85], Bailey [90], Duarte and Hayashi [86], Ogburn and Edwards [89], and Maciel et al. [87] for *Pereskia* genus and *P. guamacho*. The druse function is related to the regulation and homeostasis of calcium, as well as tolerance to stress [91], and the accumulation of soluble salts such as CaCO_3_ and CaSO_4_ in the soil [92]. 

The three species shared a common foliar anatomical characteristic: a wider adaxial epidermis than the abaxial one, a strategy for adaptation to arid environments, and high radiation intensity where a greater water reserve is ensured when there is less water availability [39]. Similarly, the xylem diameter of the three species tended to be small, which reduces hydraulic conductance and the risk of cavitation [93]. Such characteristics in xeromorphic species are common in plants in arid and semi-arid environments [92].

## 4. Materials and Methods

### 4.1. Study Area and Plant Material

This experiment was carried out on the indigenous Cerrejón Guajira Foundation farm (11°35’38.3’’ N, 72°19’29.1’’ W) in the municipality of Uribia, Department of La Guajira, Colombia (Figure A1), at the end of the wet season in November of 2019, with an average temperature of 27.7 °C, an average relative humidity of 80.0%, and a vapor pressure deficit of 0.8 KPa. Following a reconnaissance field survey that sampled 30 resource islands, the more abundant nurse tree species with foliar characteristics were chosen for measuring multiple foliar traits. The following nurse tree species that form resource islands were selected: *Haematoxylum brasiletto* with an average height of 4.8 m and a canopy cover of 45 cm^2^, *Pithecellobium dulce* with an average height of 4 m and 38 cm^2^ coverage, and *Pereskia guamacho* with an average height of 4 m and 42 cm^2^ coverage. The plants were identified with the herbarium of the University of La Guajira, Wunü’Ülia’, and one specimen of each species was kept in the herbarium for reference.

The quantification and analyses of physiological, biochemical, and anatomical variables were carried out in the laboratories of physiology and plant biochemistry, tissue culture, and optical equipment in the Departamento de Biología of the Universidad Nacional de Colombia, Bogotá, Colombia.

### 4.2. Physiological Traits

Foliar water status, leaf area, specific leaf area, leaf thickness, dry mass leaf content, and stomatal density were determined on the same leaves. Separate leaves were used for the stomatal conductance and chlorophyll a fluorescence measurements. Similarly, pH and biochemical traits were determined with different leaves. The leaves used for the quantification of the functional traits were mature, healthy, and fully expanded from the middle-third of the branch in the middle-third of the canopy in seven trees per species.

Water content (WC; %), relative water content (RWC; %), and water saturation deficit (WSD; %) were determined in three leaves per tree. Briefly, after fresh weight, leaves were soaked in sterile water for 12 h to reach saturation weight. Then, leaves were dried at 60 °C for 48 h to determine dry weight. Finally, the equations of Pérez et al. [94] were used to calculate WC, RWC, and WSD. Leaf area (LA; cm^2^) and specific leaf area (SLA; cm^2^/g) were determined based on the methodology described by Suárez Salazar et al. [95], using three leaves per tree. For LA, photographs were taken of the leaves with a white background, which were analyzed with ImageJ (Java 1.8.0_112). The SLA was determined as the ratio between the LA and the respective dry weight. Leaf thickness (mm) and dry mass leaf content (DMLC; mg/g) were calculated based on the methodology described by Salgado-Negret et al. [20], using three leaves per tree. The thickness of each leaf was determined with a digital micrometer (Fisher Scientific, Traceable™ Digital Caliper, Pittsburgh, PA, USA). 

pH determination was carried out according to the methodology proposed by Cornelissen et al. [52], with some modifications. Leaves were collected from each tree in paper bags, dried for 72 h at 60 °C to obtain 0.5 g of dry plant material, and placed in falcon tubes with 4 mL of distilled water, maintaining a volume ratio of 1:8 (P:V). Subsequently, they were placed in a horizontal agitator (DLAB Scientific Co, SK-L330-Pro, Beijing, China) at 250 rpm for 1 h and centrifuged at 6000 rpm for 15 min at 4 °C (Hettich® Universal 320/320R, GmbH & Co. KG, Wertheim, Germany). The pH was measured with a pH meter (CG 820, Schott Gerate GmbH, Hofheim, Germany), which was calibrated with the corresponding buffer solutions (pH 4 and 7).

Stomatal conductance (g_s_, mmol·m^−2^·s^−1^) was determined in three leaves per tree with a porometer (Decagon Devices Inc., Pullman, WA, USA) three times throughout the day (3:30–5:30 h, 12:30–14:30 h and 17:30–19:30 h) for the three species [96]. 

Stomatal density was determined with an epidermal impression made by coating the surface of 1 leaf with nail varnish per individual [96]. The surface morphology/anatomical characteristics were studied with an Olympus BX50 light microscope (Olympus Optical Co. Ltd., Tokyo, Japan), and the stomatal density (SD, stomata/mm^2^) was calculated in 5 optical fields at 400 magnification. For *P. guamacho*, which is an amphistomatic species, the equation proposed by Pérez et al. [94] was used to average the stomatal density of the leaves.

Chlorophyll a’s fluorescence was measured with a non-modulated fluorometer Pocket PEA (Hansatech Instruments Ltd, King’s Lynn, Norfolk, UK), in leaves that were previously adapted to darkness for 30 min. The light pulse lasted 1 s at an intensity of 3500 μmol photons·m^−2^·s^−1^. The measurements were taken predawn between 2:00 and 5:30 h [65].

Data were analyzed with chlorophyll a fluorescence rapid-phase parameters, obtained with PEA Plus Software Version 1.12 (2016): PI_ABS_, Φ_Po_, Φ_Eo_, Ψ_o_, ABS/RC, DIo/RC, TRo/RC, ETo/RC, REo/RC, ABS/CSo, DIo/CSo, TRo/CSo, ETo/CSo, and REo/CSo (Table 4). In addition, OJIP chlorophyll a fluorescence induction curves were developed for each of the nurse species with R (Version 4.1.0, R Core Team, Foundation for Statistical Computing, Vienna, Austria).

### 4.3. Biochemical Traits

For biochemical traits, a pool of mature, healthy, and fully expanded leaves was taken from the middle-third branches in the middle-third of the canopy per tree (four trees per species). Subsequently, each pool of leaves per individual was homogenized and macerated with liquid nitrogen to obtain four technical replicas for the following determinations.

#### 4.3.1. Photosynthetic Pigments

The quantification of chlorophyll a (Chl_a_), chlorophyll b (Chl_b_), total chlorophylls (Chl_total_), carotenoids ©, and the Chl_a_/Chl_b_ ratio (mg g^−1^ fresh weight—FW) was carried out on macerated leaf samples using the methodology described by Lichtenthaler [99] and Solarte et al. [100]. Determinations were carried out using 80% cold acetone under dark conditions to avoid the degradation of photosynthetic pigments.

#### 4.3.2. Proline Content

The proline content (μg proline mg^−1^ FW) was determined according to Bates et al. [101] and Moreno et al. [32]. Macerated leaf samples (0.5 g) were extracted in 5 mL of 3% (*w*/*v*) sulphosalicylic acid. The extract (1 mL) was mixed with an equivalent of acid ninhydrin reagent and glacial acetic acid, and the mixture was boiled for 1 h. After cooling in an ice-water bath for 15 min, toluene (3 mL) was added and mixed. The absorbance of the organic phase was read at 520 nm, using toluene as a blank. The concentration was determined from the L-proline standard curve.

#### 4.3.3. Total Sugar Content

Soluble sugars (μg Total sugar content mg^−1^ FW) were extracted using the methodology described by Dubois et al. [102] and Moreno et al. [32]. Macerated leaf samples (0.2 g) were mixed with distilled water (5 mL), shaken for 1 h, and centrifugated at 6000 rpm for 30 min. Then, 30 μL of supernatant was removed and mixed with 180 μL of water, 200 μL of 80% phenol, and 1 mL of 98% sulfuric acid. A standard curve was made using D-glucose to quantify total sugars, which were measured at an absorbance of 490 nm.

### 4.4. Anatomical Characteristics

Leaves were taken from the middle-third branches in the middle-third of the canopy per tree (four trees per species) and preserved in 70% alcohol for subsequent processing. The leaves were fixed in FAA (formaldehyde: acetic acid: 70% ethanol, 10:5:85), stored in 70% ethanol, and subsequently treated following the protocol of Johansen [103], as modified by Robles et al. [104]. This protocol involved standard dehydration methods using a clearing agent (Histoclear), paraffin infiltration, sectioning with a rotary microtome, and placement on microscope slides. The slides were stained with astra-blue and basic fuchsin and deposited. They were analyzed and photographed with an Olympus BX-50 in the optical equipment laboratory in the Departamento de Biología, Universidad Nacional de Colombia. Digital images were processed and edited with the program ImageJ (Java 1.8.0_112).

The images were used for the foliar anatomical description of the three nurse tree species, to classify the type of stomata [105], to classify the shape of the stomata [106], and to describe the stomatal structure [107]. Additionally, measurements were taken in leaf cross-sections for the thickness of the mesophyll, the superior and inferior epidermis, the spongy and palisade parenchyma, the cuticle, and the central rib. The diameter of vessels of the xylem, the width of the wall of the vessels, the length of the xylem within the central rib, and the percentage that represents the length of the xylem with respect to the width of the midrib were measured. The measurements were taken with ImageJ (Java 1.8.0_112). Each variable was determined with 3 leaves per species. A total of 5 cuts were taken from the middle part of each leaf (*n* = 15).

### 4.5. Statistical Analysis

The physiological and anatomical traits were characterized with a summary of the variables for each species (median ± IQR), and the respective Wilcoxon range tests were performed in pairs with Bonferroni correction to determine statistically significant differences between the species (α = 0.05). Chlorophyll a fluorescence data for each species were shown in spider diagrams, representing the medians of the parameters. The data obtained in the first second of the rapid phase of fluorescence emission of chlorophyll a for each of the species were used for the OJIP curve (logarithmic scale). For the biochemical traits, the results were presented in bar diagrams (median ± IQR), and statistical differences between the species were determined with their respective Wilcoxon tests in pairs with Bonferroni correction (α = 0.05). The percentages were calculated with angular transformation for the test. Figures and statistical tests were performed with R Version 4.1.0 (2021).

## 5. Conclusions

This is the first study integrating physiological, biochemical, and anatomical leaf traits in nurse trees from resource islands. Our results indicate that *H. brasiletto* has a sclerophyllous leaf and performs stomatal control to reduce water loss and improve water use efficiency. In addition, it has high photochemical efficiencies and a high proline content, suggesting a cellular osmoregulation strategy. *P. dulce* has a sclerophyll type leaf and does not perform stomatal closure, possibly resulting in better photosynthetic performance because of constant CO_2_ input. It also had the highest Fm, which was evidenced in the OJIP curve and energy flux parameters, indicating that this species has high photochemical efficiency. Finally, *P. guamacho* has foliar and anatomical characteristics that are typical of succulent species, allowing the leaves to store large amounts of water. Additionally, it performs stomatal control and has a low photochemical efficiency that is related to excess energy dissipation. These species have strategies that allow them to live in semi-arid environments with high temperatures, high radiation, and low water availability.

## Figures and Tables

**Figure 1 plants-11-01723-f001:**
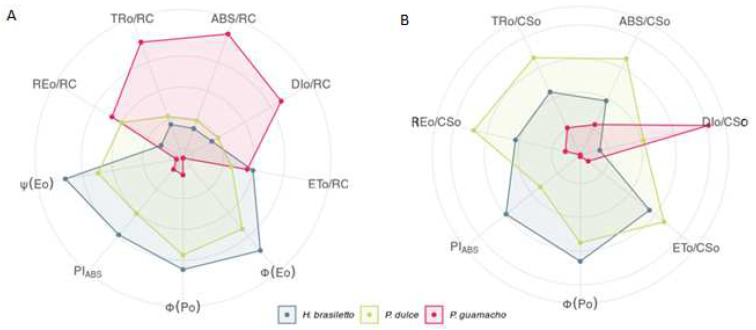
Spider-plot showing various parameters obtained from the rapid emission phase of chlorophyll a fluorescence in *H. brasiletto, P. dulce*, and *P. guamacho*. Each line represents the average of 7 plants. (**A**) Specific energy flux parameters (ABS/RC, DIo/RC, Tro/RC, Eto/RC, and REo/RC), quantum efficiencies (Φ_Po_, Φ_Eo_, and Ψ_o_), and performance index (PI_ABS_). (**B**) Phenomenological energy flux parameters (ABS/CSo, Dio/Cso, Tro/Cso, Eto/Cso, and REo/CSo).

**Figure 2 plants-11-01723-f002:**
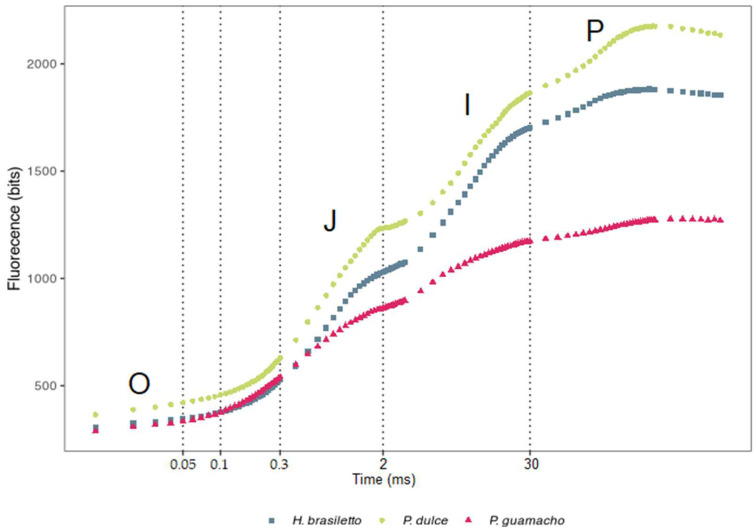
The fast fluorescence rise (measured with the PEA instrument) in *H. brasiletto, P. dulce*, and *P. guamacho* leaves. Transient curves of each line represent the average of 7 plants.

**Figure 3 plants-11-01723-f003:**
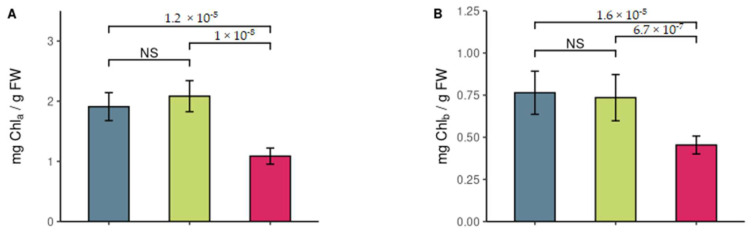
Biochemical traits: (**A**) chlorophyll a (Chl_a_); (**B**) chlorophyll b (Chl_b_); (**C**) total chlorophylls (Chl_total_); (**D**) carotenoid content (**C**); (**E**) Chl_a_/Chl_b_ ratio; (**F**) proline content; (**G**) total sugar content. Contents measured in the nurse tree species *H. brasiletto, P. dulce*, and *P. guamacho.* Values present median ± IQR (*n* = 4). The numbers in the upper brackets indicate the *p*-value, according to the Wilcoxon test for paired species with Bonferroni correction (NS: not significant).

**Figure 4 plants-11-01723-f004:**
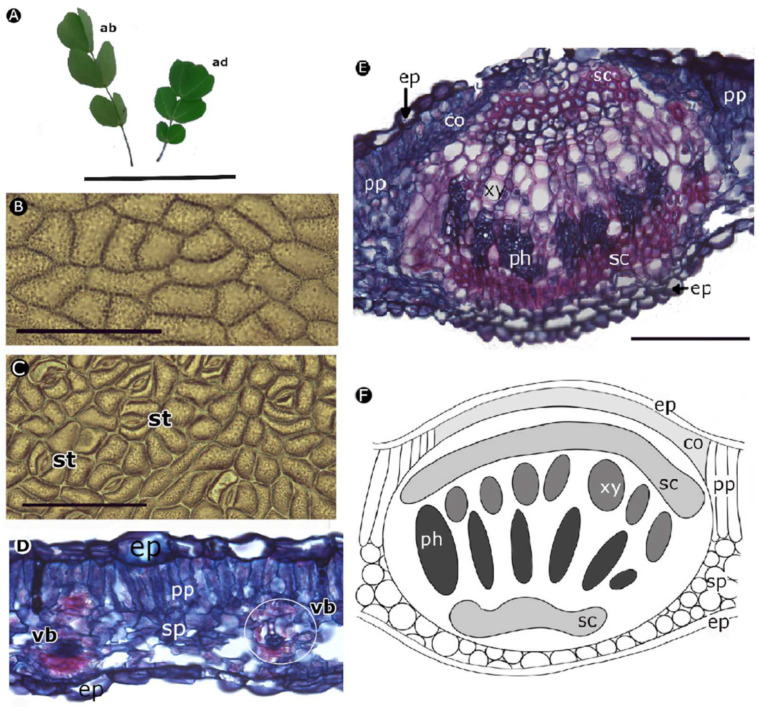
Morpho-anatomy of leaves of *Haematoxylum brasiletto* [(**D**,**E**): optical microscopy, sections stained with basic fuchsin and astra blue]. (**A**) Leaves; (**B**) adaxial side; (**C**) abaxial side; (**D**): foliar blade; (**E**): midrib; (**F**): diagram of midrib organization and part of leaf blade (ab: abaxial; ad: adaxial; co: collenchyma; ep: epidermis; ph: phloem; pp: palisade parenchyma; sp: spongy parenchyma; sc: sclerenchyma; st: stomata; xy: xylem; vb: vascular bundle). Scale bars: (**A**) = 5 cm; (**B**–**E**) = 0.1 mm.

**Figure 5 plants-11-01723-f005:**
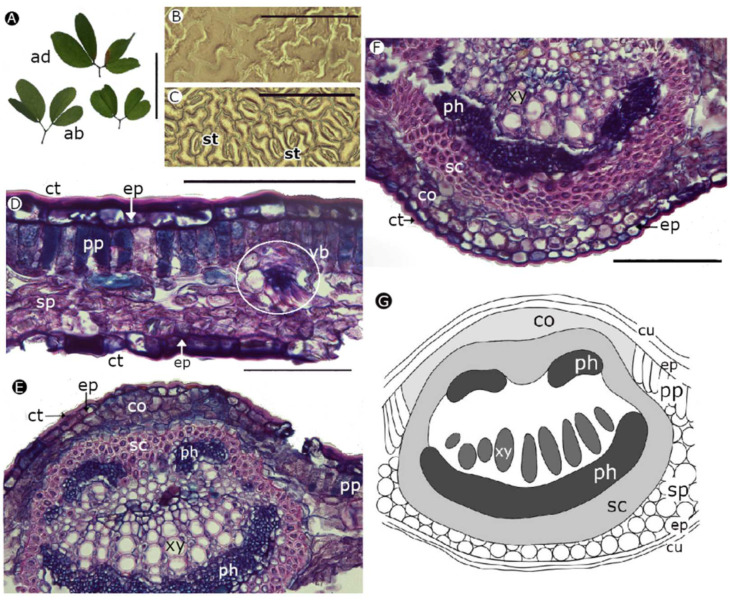
Morpho-anatomy of leaves of *Pithecellobium dulce* [(**D**–**F**): optical microscopy, sections stained with basic fuchsin and astra blue]. (**A**) Leaves, (**B**) adaxial side, (**C**) abaxial side, (**D**) foliar blade, (**E**) and (**F**) midrib, and (**G**) diagram of midrib organization and part of leaf blade. (ab: abaxial; ad: adaxial; co: collenchyma; ct: cuticle; ep: epidermis; ph: phloem; pp: palisade parenchyma; sp: spongy parenchyma; sc: sclerenchyma; st: stomata; xy: xylem; vb: vascular bundle). Scale bars: (**A**) = 5 cm; (**B**–**F**) = 0.1 mm.

**Figure 6 plants-11-01723-f006:**
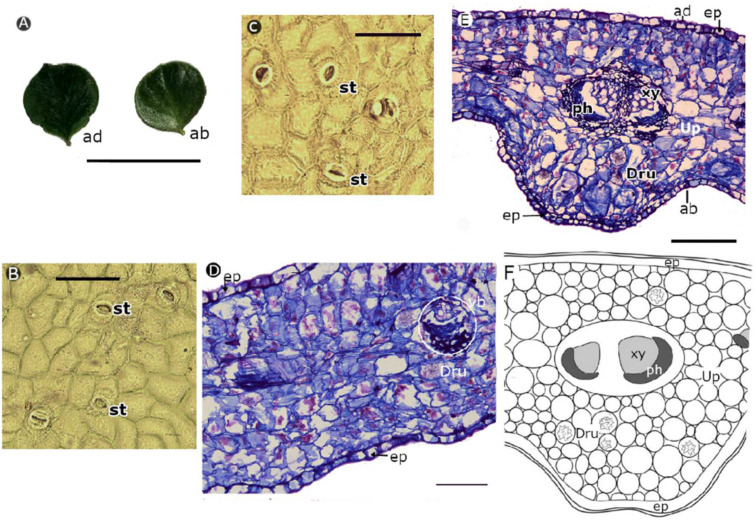
Morpho-anatomy of leaves of *Pereskia guamacho* [(**C**–**E**): optical microscopy, sections stained with basic fuchsin and astra blue]. (**A**) Leaves, (**B**) adaxial side, (**C**) abaxial side, (**D**) foliar blade, (**E**) midrib, and (**F**) diagram of midrib organization. (ab: abaxial; ad: adaxial; dru: druses; ep: epidermis; ph: phloem; st: stomata; xy: xylem; up: undifferentiated parenchyma; vb: vascular bundle). Scale bars: (**A**) = 5 cm; (**B**–**D**) = 0.1 mm; (**E**) = 0.2 mm.

**Table 1 plants-11-01723-t001:** Functional traits in leaves of *H. brasiletto*, *P. dulce*, and *P. guamacho*.

Variable	*H. brasiletto*	*P. dulce*	*P. guamacho*	Group
WC (%)	151.79 ± 10.1	121.43 ± 34.87	744.59 ± 87.76	(a, b, c)
RWC (%)	83.46 ± 10.48	90.85 ± 4.21	97.78 ± 0.61	(a, b, c)
WSD (%)	16.54 ± 10.49	9.15 ± 4.21	2.22 ± 0.6	(a, b, c)
LA (cm^2^)	3.67 ± 1.56	6.91 ± 3.7	7.39 ± 1.96	(a, b, b)
SLA (cm^2^/g)	134.19 ± 10.88	108.88 ± 23.82	113.58 ± 38.79	(a, b, b)
LDMC (mg/g)	364.8 ± 19.97	437.47 ± 21.63	115.38 ± 13.28	(a, b, c)
Thickness (mm)	0.2 ± 0.03	0.23 ± 0.05	0.81 ± 0.34	(a, b, c)
pH	5.17 ± 0.27	5.72 ± 0.08	5.24 ± 0.32	(a, b, a)
SD (stomata/mm^2^)	223.53 ± 55.88	647.06 ± 164.7	38.06 ± 6.36	(a, b, c)
g_s_ 3:30–5:30 h (mmol·m^−2^·s^−1^)	64.8 ± 20	60.6 ± 20.7	50.8 ± 24.3	(a, a, a)
g_s_ 12:30–15:00 h (mmol·m^−2^·s^−1^)	45.3 ± 16.55	87.6 ± 14.9	22 ± 3.3	(a, b, c)
g_s_ 17:30–19:30 h (mmol·m^−2^·s^−1^)	40.6 ± 5.5	39.8 ± 6.6	35.9 ± 6.9	(a, ab, c)

WC: water content; RWC: relative water content; WSD: water saturation deficit; LA: leaf area; SLA: specific leaf area; LDMC: leaf dry mass content. Thickness; pH; SD: stomatal density; g_s_: stomatal conductance three times throughout the day (3:30–5:30 h, 12:30–15:00 h, and 17:30–19:30 h).

**Table 2 plants-11-01723-t002:** Parameters obtained from the rapid emission phase of chlorophyll a fluorescence.

Variable	*H. brasiletto*	*P. dulce*	*P. guamacho*	Group
PI_ABS_	5.84 ± 2.19	4.42 ± 3.37	1.16 ± 0.55	(a, a, b)
Φ_Po_ (Fv/Fm)	0.84 ± 0.01	0.82 ± 0.02	0.73 ± 0.09	(a, a, b)
Φ_Eo_	0.48 ± 0.05	0.44 ± 0.09	0.3 ± 0.05	(a, b, c)
Ψ_Eo_	0.57 ± 0.07	0.53 ± 0.09	0.43 ± 0.03	(a, b, c)
ABS/RC	1.19 ± 0.22	1.24 ± 0.34	1.86 ± 0.16	(a, a, b)
DIo/RC	0.19 ± 0.03	0.22 ± 0.09	0.5 ± 0.22	(a, a, b)
TRo/RC	1 ± 0.19	1.02 ± 0.26	1.35 ± 0.12	(a, a, b)
ETo/RC	0.56 ± 0.05	0.53 ± 0.06	0.56 ± 0.11	(ab, b, a)
REo/RC	0.14 ± 0.03	0.15 ± 0.03	0.17 ± 0.02	(a, b, b)
ABS/CSo	309 ± 58	360 ± 101	290 ± 108	(a, b, a)
DIo/CSo	51.45 ± 9.96	64.24 ± 22.97	84.74 ± 25.94	(a, b, c)
TRo/CSo	259.78 ± 48.07	296.93 ± 72.13	224.88 ± 106.8	(a, a, b)
ETo/CSo	147.75 ± 31.07	159.42 ± 48.64	98.85 ± 36.06	(a, a, b)
REo/CSo	37.27 ± 14.76	45.53 ± 13.11	27.25 ± 6.87	(a, b, c)

Performance index (PI_ABS_), quantum efficiencies (Φ_Po_, Φ_Eo_, Ψ_o_), specific energy flux parameters (ABS/RC, DIo/RC, TRo/RC, ETo/RC, and REo/RC), and phenomenological energy flux parameters (ABS/CSo, DIo/CSo, TRo/CSo, ETo/CSo, and REo/CSo).

**Table 3 plants-11-01723-t003:** Anatomical measurements of the leaf blade of the nurse tree species *H. brasiletto, P. dulce*, and *P. guamacho*.

Variable	*H. brasiletto*	*P. dulce*	*P. guamacho*	*p*-Value	Groups
Mesophyll (mm)	0.103 ± 0.003	0.092 ± 0.01	0.415 ± 0.149	6 × 10^−8^	(a, b, c)
Upper epidermis(mm)	0.016 ± 0.003	0.01 ± 0.002	0.022 ± 0.003	2 × 10^−7^	(a, b, c)
Lower epidermis (mm)	0.012 ± 0.003	0.01 ± 0.001	0.021 ± 0.005	1 × 10^−7^	(a, a, b)
Palisade parenchyma (mm)	0.033 ± 0.006	0.031 ± 0.013	-	NS	(a, a, -)
Spongy parenchyma (mm)	0.04 ± 0.006	0.035 ± 0.006	-	3 × 10^−3^	(a, b, -)
Undifferentiated parenchyma (mm)	-	-	0.368 ± 0.139	-	-
Upper cuticle (mm)	-	0.002 ± 0	-	-	-
Lower cuticle (mm)	-	0.001 ± 0.001	-	-	-
IDV (mm)	0.012 ± 0.002	0.011 ± 0.001	0.011 ± 0.004	NS	(a, a, a)
WWV (mm)	0.002 ± 0	0.002 ± 0	0.003 ± 0.001	4 × 10^−5^	(a, ab, c)
LX (mm)	0.072 ± 0.004	0.053 ± 0.007	0.049 ± 0.016	4 × 10^−5^	(a, b, b)
Percentage (%)	27.613 ± 3.517	19.203 ± 7.631	43.023 ± 12.075	4 × 10^−5^	(a, b, c)

Internal diameter of vessels of the xylem (IDV), width of the wall of the vessels (WWV), length of the xylem within the midrib (LX), and the percentage that represents the length of the xylem with respect to the width of the midrib (Percentage).

**Table 4 plants-11-01723-t004:** Formulae and glossary of chlorophyll fluorescence parameters derived from OJIP test, as described by Brestic and Zivcak [97], Gururani et al. [98], Strasser et al. [30] and González Moreno [26].

Parameter	Basic Physiological Interpretation	Equation
Basic parameters derived from OJIP transient		
F_o_	Minimum fluorescence, when all RC PS II are open	-
F_m_	Maximum fluorescence, when all RC PS II are closed	-
F_v_	Variable fluorescence	Fv = Fm − Fo
V_t_	Relative variable fluorescence at time t	$V_{t}=\frac{\mathrm{Ft}-\mathrm{Fo}}{\mathrm{Fm}-\mathrm{Fo}}$
V_j_	Relative variable fluorescence at time 2 ms (J-step)	$V_{J}=\left(F_{2\;\mathrm{ms}}-\mathrm{Fo}\right)/($ Fm − Fo)
V_i_	Relative variable fluorescence at time 30 ms (I-step)	$V_{I}=\left(F_{30\;\mathrm{ms}}-\mathrm{Fo}\right)/($ Fm − Fo)
TF_m_	Time of reaching maximum fluorescence	tf max
M_o_	Initial slope of relative variable chlorophyll fluorescence, express the rate of electron trapping	$M_{o}=\frac{\mathrm{TRo}}{\mathrm{RC}}-\frac{\mathrm{ETo}}{\mathrm{RC}}$ $=4*\frac{\left(F_{300\;\mu s}-\mathrm{Fo}\right)}{\left(\mathrm{Fm}-\mathrm{Fo}\right)}$
**Quantum yields**		
$\varphi_{Po}$	Quantum efficiency of photosystem II	$\varphi_{\mathrm{Po}}={\mathrm{TR}}_{O}/\mathrm{ABS}=\mathrm{Fv}/\mathrm{Fm}$
$\varphi_{Eo}$	Probability that an absorbed exciton moves an electron after Q_A_^−^	$\varphi_{\mathrm{Eo}}=\left(1-\mathrm{Fo}/\mathrm{Fm}\right)\left(1-V_{j}\right)$
$\psi_{o}$	Efficiency with which a trapped exciton can move an electron after Q_A_^−^	$\psi_{o}={\mathrm{ET}}_{O}/{\mathrm{TR}}_{O}=\left(1-V_{j}\right)$
$\delta_{Ro}$	Efficiency with which an electron from the intersystem electron carriers moves to reduce end electron acceptors at the PSI acceptor side (RE)	$\delta_{\mathrm{Ro}}=\mathrm{REo}/\mathrm{ETo}$ $=\left(1-V_{i}\right)/\left(1-V_{j}\right)$
**Specific energy fluxes (per reaction center—RC)**		
ABS/RC	Absorption (ABS) per RC	$\frac{\mathrm{ABS}}{\mathrm{RC}}=\left(M_{o}\;/{\;V}_{j}\right)\;/\;\left[\;1-\left(\mathrm{Fo}\;/\;\mathrm{Fm}\right)\right]$
DIo/RC	Dissipation (DI) at time 0 per RC	DIo/RC =(ABS/RC)−(TRo/RC)
TRo/RC	Trapped energy flux (TR) at time 0 per RC	$\frac{\mathrm{TRo}}{\mathrm{RC}}=\varphi_{\mathrm{Po}}*\left(\frac{\mathrm{ABS}}{\mathrm{RC}}\right)$
ETo/RC	Electron transport flux (ET) at time 0 per RC	$\frac{\mathrm{ETo}}{\mathrm{RC}}=\psi_{o}*\left(\frac{\mathrm{TRo}}{\mathrm{RC}}\right)$
REo/RC	Reduction in final electron acceptors on the electron acceptor side of PSI (RE) at time 0 per RC	$\frac{{\mathrm{RE}}_{O}}{\mathrm{RC}}=M_{o}*\left(1/V_{j}\right){\;*\;\psi }_{\mathrm{Eo}}{\;*\;\delta }_{\mathrm{Ro}}$
**Phenomenological fluxes (per excited cross-section—CS)**		
ABS/CSo	Absorption (ABS) per CS using the F_o_^dark^ value the samples exhibited while in a dark-adapted state	$\frac{\mathrm{ABS}}{\mathrm{CSo}}=F_{o}{}^{\mathrm{dark}}$
DIo/CSo	$\mathrm{Dissipation}\;(\mathrm{DI})\;\mathrm{at}\;\mathrm{time}\;0\;\mathrm{per}\;\mathrm{CS}\;\mathrm{when}\;\mathrm{all}\;\mathrm{reaction}\;\mathrm{centers}\;\mathrm{are}\;\mathrm{open}\;(F_{o}{}^{\mathrm{dark}})$	DIo/CSo = (ABS/CSo) − (TRo/CSo)
TRo/CSo	Trapping (TR) at time 0 per CS	$\mathrm{TRo}/\mathrm{CSo}=\varphi_{\mathrm{Po}*}$ (ABS/CSo)
**ETo/CSo**	Electron transport (ET) at time 0 per CS	$\mathrm{ETo}/\mathrm{CSo}=\varphi_{\mathrm{Po}}$$\;*\;\psi_{o}\;*$ (ABS/CSo)
**REo/CSo**	Reduction in final electron acceptors on the electron acceptor side of PS I (RE) at time 0 per CS	$\mathrm{REo}/\mathrm{CSo}=\varphi_{\mathrm{Po}}\;*\;$$\psi_{o}$$\;*\;\delta_{\mathrm{Ro}}\;*$ (ABS/CSo)
**Performance index**		
**PI_ABS_**	Performance index	PI_ABS_ = [RC/ABS][φ_PO_/(1 − φ)] ∗ [ψ/1 − ψ_o_]

## Data Availability

Not applicable.

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
