# Peer review of "Foliar Functional Traits of Resource Island-Forming Nurse Tree Species from a Semi-Arid Ecosystem of La Guajira, Colombia"

_plants, 2022, doi:10.3390/plants11131723_

Round 1
Reviewer 1 Report
Semi-arid enviroments is very important for the globe ecosystem. The resource islands are very important for the semi-arid vegetation. So the nurse species have some special strategy to fit the enviroment is very interesting. This study is very important. However, the nurse species in the semi-arid enviroments is not only three species. Why did you choose these three species?
The character of the different species are a lot, why do you choose those character for your study?
Cerrejón Guajira Foundation farm is very unfamilier with most of the reader, maybe you can show some vegetation photos for the reader to undersand the semi-arid enviroment.
Reviewer 2 Report
Generally manuscript is well written, however, Authors should do some changes. Authors have to explain why they chose these particular species. Why these genera, these families? Pereskia belongs to Cactaceae, therefore the result of having succulent leaves in this genus is not surprising, and it is well documented in the literature (see e.g. Comparative Anatomy of the Leaf-Bearing Cactaceae - https://www.jstor.org/stable/43782467?seq=1 and also http://old.scielo.br/scielo.php?pid=S0102-695X2005000200006&script=sci_abstract&tlng=en
Could Authors give midrib organization diagrams of studied species this will be helpful for reader (see for example http://old.scielo.br/pdf/rbfar/v15n2/v15n2a06.pdf -Fig. 4)
Could Authors provide SEM documentation of leaf surfaces?
The authors should discuss the anatomical and morphological results of the species they have studied with other species of these genera. Whether there are significant differences ?
Round 2
Reviewer 2 Report
Fagua Alvarez Flórez and team corrected the manuscript responding to all suggestions from reviewers. In this form, the manuscript is clearer to the reader. Thank you for bringing in the diagrams that have improved the value of the work.
